# Viral Infections and Interferons in the Development of Obesity

**DOI:** 10.3390/biom9110726

**Published:** 2019-11-12

**Authors:** Yun Tian, Jordan Jennings, Yuanying Gong, Yongming Sang

**Affiliations:** Department of Agricultural and Environmental Sciences, College of Agriculture, Tennessee State University, 3500 John A. Merritt Boulevard, Nashville, TN 37209, USA; ytian@my.tnstate.edu (Y.T.); ygong@tnstate.edu (Y.G.)

**Keywords:** persistent viral infection, interferon, lipid metabolism, obesity

## Abstract

Obesity is now a prevalent disease worldwide and has a multi-factorial etiology. Several viruses or virus-like agents including members of adenoviridae, herpesviridae, slow virus (prion), and hepatitides, have been associated with obesity; meanwhile obese patients are shown to be more susceptible to viral infections such as during influenza and dengue epidemics. We examined the co-factorial role of viral infections, particularly of the persistent cases, in synergy with high-fat diet in induction of obesity. Antiviral interferons (IFNs), as key immune regulators against viral infections and in autoimmunity, emerge to be a pivotal player in the regulation of adipogenesis. In this review, we examine the recent evidence indicating that gut microbiota uphold intrinsic IFN signaling, which is extensively involved in the regulation of lipid metabolism. However, the prolonged IFN responses during persistent viral infections and obesogenesis comprise reciprocal causality between virus susceptibility and obesity. Furthermore, some IFN subtypes have shown therapeutic potency in their anti-inflammation and anti-obesity activity.

## 1. Viral Infectobesity: The Association of Chronic Viral Infections with Obesity

According to recent statistics, the prevalence of obesity was 39.8% and affected about 93.3 million US adults in 2015~2016 [1]. Similarly, in many other countries worldwide, overweight and obesity are also dramatically on the rise, particularly in urban areas [2]. We face an obesity epidemic that has a complex and multifactorial etiology [3]. While obesity is multifactorial, a simplified thermo-dynamic model, rather than a biological model, has prevailed in obesity clinics for over 90 years [3]. A key suggestion of this simplified model is to cut down one’s calorie intake and exercise more. While addressing proximate causes of obesity serves as a global health priority, it seems incomplete or ineffective to control obesity given that the epidemic has continued to worsen at the global level [1,2,3]. Recent investigations postulate that both physics and biology have tremendous utility for understanding obesity. In this regard, we review the biological complex of obesity and interpret its etiology from the interaction between immune and metabolic systems (i.e., immunometabolism). As pointed out by Hotamisligil [4], metabolic and immune systems are among the most fundamental requirements and are highly inter-dependent for species survival. Sequentially, obesity displays itself as a metabolic overload (shown by excess adipose tissue) in addition to an immune disorder, which accompanies a low-grade inflammatory state known as “meta-inflammation” involving both reprogrammed immune cells and adipocytes [4]. Immunologically, chronic inflammation accompanies persistent infections (either locally or systemically) as well as microbiota dysbiosis [5,6,7,8,9,10,11,12]. In addition, persistent infections could also retain energy storage via adipogenesis at adipose depots for immune preparation [5,13,14]. The visceral adipose depots in mammals, as counterparts of the insect fat body which functions as both a storage and an immune organ, may evolutionarily enhance their role in anabolic storage for combat readiness when facing long-term defense risks posed by pathogenic persistence [4,11,12,13,14]. In this context, recent studies have associated obesity with numerous microbial agents in both humans and animal models—a phenomenon described as infectobesity [15,16,17]. Microbial infections generally beget inflammation and alter metabolic function in surrounding cells or even systemically in the animal body. In this review, we recap the evidence of viral agents in association with obesity and relevant metabolic syndrome [13,14,15,16,17]. In addition to examination of the major viruses or virus-like agents (VLA) that have been associated with obesity, we emphasize shared immune responses underlying the viral infectobesity. With this regard, interferons (IFNs), as a group of crucial cytokines in regulation of antiviral and autoimmune processes, are key to understanding the pathogenesis of persistent viral infections that are associated with most, if not all, viral infectobesity [11,12,18,19]. In addition, the effect of IFNs and IFN-mediated signaling pathway (collectively known as the IFN system) in reprograming cellular lipid metabolism provides a biochemical mechanism to understand the interaction of viral infection and obesity [4,5,11,19].

As shown in Table 1, the major viruses and VLA that have been associated with obesity in both humans and animal models include members of adenoviridae, herpesviridae, phages, transmissible spongiform encephalopathies (slow virus), and hepatitides [15,16,17]. The additions to this list also include Dengue fever virus (DENV) and HIV (which belong to Flavivirade and retroviridae, respectively) in patients post the viral infections entering into a latent phase [20,21,22,23,24]. Voss and Dhurandhar (2017) rationalized the biological plausibility of infectobesity caused by these viruses [15], which included: (1) direct roles of some viruses to reprogram host metabolism toward a more lipogenic and adipogenic status; (2) the probability that humans may exchange microbiota components (emphasizing its viral component, i.e., virome/virobiota) from livestock reservoirs that have been aggressively selected for efficient weight/fat gain; and (3) the adaptation of host immune and metabolic system under persistent viral infections [15,16,17,18]. With regard to these, we emphasize the sequential immuno-metabolic adaption to persistent viral infections, especially the subverted IFN response, which emerges as a key drive for adipogenesis and obesity-associated immune suppression. This, in turn, underlies reciprocal causality between obesity and higher susceptibility of obese individuals to viral infections [11,12,13,14,15,16].

To recap the evidence of viral infectobesity, adenoviruses (Ad) including an avian isolate from India (SMAM-1) and several serotypes of human Ad have been associated with obesity regarding epidemiological correlation in certain subsets of humans and animals [15,24]. In addition, as SMAM-1 infection caused fat gain and high body mass index (BMI) in both birds and probably humans. At least five human Ad serotypes including Ad-5, -9, 31, -36, and -37 have been observed to cause obesity at different levels. First, these five Ad serotypes induced adipocyte differentiation and increased lipid synthesis in cultured animal fibroblasts or pre-adipocytes. Second, Ad-5, Ad-37, and particularly Ad-36 infections increased the adipose index in animal models including chickens (Ad-36 and -37), mice (Ad-5 and -36), rats, and marmosets (Ad-36). Third, Ad-5 has been associated with child obesity and Ad-36 in both adults and children [15,24]. Mechanistically, Ad-36 infection activates peroxisome proliferator activated receptor-γ (PPARγ) (which is the so-called master-switch of adipocyte development) to signal adult stem cells to become adipocytes; meanwhile, the virus also induces the expression of some cell glucose transporters to facilitate energy supply [25,26]. In addition, adenoviral E4orf1 protein (early region 4 open reading frame 1) was identified as a viral mechanism mediating acute adipogenesis in cells and animals [27,28]. Similarly, all other viruses and VLA listed in Table 1 have also been associated with obesity in animals and humans through pathological studies or epidemiological observations; however, other viral mechanisms (such as viral factors regulating inflammatory response and lipogenesis in adipose depots) directly engaged in adipogenic effects warrant further investigation [15,29,30,31,32,33,34,35,36,37,38,39,40,41,42,43,44,45]. Notably, because phages do not directly infect animal cells, these bacterial viruses plausibly exert an adipogenic effect by affecting the symbiosis of microbiota [31,32,33,34]. The adipogenic observation in the patients of prion diseases (or coined as “slow viruses” previously), is plausibly ascribed to host immuno-metabolic adaptation to the chronic pathogenesis during disease progress, because no adipogenic effect has been associated to the prion protein [15,37,38,39,40,41,42]. In this context, we emphasize that host immuno-metabolic adaptation to prolonged viral infection comprises a common etiology for viral infectobesity. As multiple immune factors, particularly inflammatory cytokines (such as IL-1, IL-6, and TNF-α) and adipokines (such as leptin and adiponectin), have been well-addressed elsewhere in obesity, we highlight the regulatory role of the IFN system in immunometabolic regulation in the scenario of viral infectobesity [4,11,12,30].

## 2. IFNs Emerging as a Key Factor to Mediate Viral Persistence and Relevant Infectobesity 

Interferons (IFN) are vital antiviral cytokines evolved in vertebrates. Host cells secrete three types of IFN peptides (i.e., type I, II and III IFNs) in response to viral infections. Three types of IFNs not only differ in their molecular signatures and bind to distinct cell receptors, but also play distinct tissue/cell-specific roles in control of viral infections. In brief, the innate immune IFNs, which include multiple subtypes of type I and type III IFNs, provide early and local protection during viral entry into animal portals. In contrast, IFN-γ, as the only member of type II IFN, is primarily produced by specific immune cells (including natural killer cells (NK) and activated T cells) to mediate adaptive immunity for eliminating dangerous pathogens that evade innate immunity [11,12,46,47,48,49]. Persistent viral infections result from the virus escaping from immune clearance and entering into a co-existing stage within the host cells to avoid excessive damage. The mechanisms to maintain persistent viral infections are incompletely understood, but believably involve both modulation of viral actions and reprograming of the host metabolic and immune responses [11,12,49]. Previous studies in mouse models proposed that ineffective adaptive immune responses including lack of effective cytotoxic T cells (CTL) and neutralizing antibodies are needed for viral persistence [11,12]. However, recent studies indicate that innate immune IFNs are necessary in the regulation of viral infections during both early and persistent phases [11,12,46,47,48,49]. Such as in mice infected by norovirus (MNV) and lymphocytic choriomeningitis virus (LCMV), IFN-α/β played a systemic role in control of the viral replication and clearance; and IFN-λs were effective in clearing enteric persistence of MNV [12,49,50]. Therefore, innate immune IFNs preside over antiviral immunity to intervene chronic viral infection through mediating at both innate (stimulation of expression of a plethora of ISG expression) and adaptive immune levels (exerting either positive or negative impacts on CTL and B-cell functions) [49,50,51]. Numerous studies in HCV, HIV-1, and simian immunodeficiency virus (SIV) as well as multiple viruses in livestock species place sustained type I IFN expression as a common nexus in the pathogenesis of multiple chronic diseases [11,12,49]. Regarding viral pathogenesis, IFN responses during the acute phase of viral infections is immuno-stimulatory and antiviral effective; however, the prolonged IFN production (particularly of type I IFNs) in response to viral persistence is emerging as a double-edge sword actually driving inflammation and immunosuppression [12,49,50,51]. In this context, examination of reported studies about viral infectobesity indicates that most, if not all, of these obese situations correlate to the chronic phases or persistent stages of viral infections (Table 1) [11,12,13,14,15,16,17]. In addition, the antagonism to the host IFN system comprises a major mechanism of viruses that are capable of being persistent or even harnessed as components of the host microbiota (virobiota) [52,53,54,55,56,57,58,59,60,61,62]. In other words, the prolonged IFN response observed during most chronic viral pathogeneses may underlie, at least partly, the immune dysfunction [11,12] that triggers metabolic disorders and obesity (Table 1) [4,13,14,15,16,17,63].

## 3. Gut Microbiota and Intrinsic Interferon Response 

Current studies link gut microbiota to the development of obesity and its related comorbidities in the following aspects [64,65,66,67]. First, changes of gut microflora affect adipogenesis or the pathogenesis of obesity. It has been shown that healthy individuals have highly diverse gut microbiota than those with obesity. The association of some bacterial taxonomic groups with obesity and relevant comorbidities was observed such as lower content of Bacteroidetes in obese people and the association of Lactobacillus and Clostridium with insulin resistance. These findings indicate that the disorder of gut microflora could be a drive to obesity development. Second, germ-free (GF) animal models provide eloquent evidence to address the role of gut microbiota in obesity induction. Without gut microbiota colonization, GF mice were highly resistant to obesity induced by a high-fat diet than normal mice despite high food intakes. Furthermore, GF mice that received fecal microbiota transplantation (FMT) from an obese donor had higher weight gain than those receiving it from a lean donor. Misuses and overuses of antibiotics in the animal industry and human medicine impact on gut microbial ecology, leading to antibiotic resistance. Relevant studies using sub-therapeutic antibiotic treatment indicated that it selectively eliminated vulnerable taxonomic groups of the gut microbiome, thus altering microbiome composition that results in elevated hepatic metabolism of lipids and cholesterol in young mice [64,65,66,67]. Mechanistically, gut microbiota affect obesity development in an inter-systemic manner. Locally in the gut, obesity-associated gut microbiota were selected for three properties: (1) a more efficient harvest of energy from the diet; (2) gene reprogramming to facilitate energy anabolism; and (3) secretion of bioactive molecules and hormones to affect host appetite. The decrease in gut microbiota diversity in obese individuals impaired the intestinal barrier function and disturbed immune homeostasis by inducing low-grade inflammation, which was considered an early trigger for adipogenesis [64,65]. Along with microbiome decomposition, the microbial shedding molecules such as liposaccharides (LPS) induce further gut inflammation and immune-cell infiltration. Short-chain fatty acid (SCFAs) released by microbiota directly fuel insulin-mediated lipogenesis and promote adipocyte differentiation. Beyond the gut, the dysbiosis of gut microbiota also signals inflammation and hepatic T cell infiltration to manifest as fatty liver frequently associated with obesity. Gut microbiota can also directly signal gut afferent neurons and mediate the secretion of appetite hormones to regulate host metabolism through the gut–brain axis and an endocrine pathway [66,67]. Therefore, collected evidence indicates that local alteration in gut microbiota will lead to systemic reprograming in energy and lipid metabolism, immune response, and endocrine functions that culminate in obesogenesis [64,65,66,67].

However, a majority of current studies about microbiota focus on cellular microbiome (including bacteria, archaea, fungi, and protozoa), but overlook the viral community (i.e., virobiota or virome) where it is at least ten-fold over more numerous than its cellular counterpart [68,69,70,71]. Indeed, a human individual regardless of health status contains at least a dozen taxa of eukaryotic viruses and bacterial phages. While phages potentially affect obesity indirectly through modulating bacterial composition, eukaryotic viruses can directly interact with host cells to result in an ideal commensalism (symbiosis) or cause either symptomatic or asymptomatic pathogenesis (dysbiosis). Compared with wild-type mice, significantly higher viral contents were detected in the feces of obese mice. It was also indicated that increased viral content, particularly viral RNA, was tightly correlated with an increase of fat mass and potential hyperglycemia. Only total viral DNA/RNA mass was compared in this study, but the viral taxonomic classification was not performed using metagenomic approaches [70]. However, their data indicated that this obesity-correlated increase of viral content was mostly ascribed to phages [70]. This reinforces the points from several recent reviews that the gut virome is a missing link underlying human diseases like cancers and obesity [68,69,71]. Further studies are needed to decipher the composition of gut commensal viruses and their role in obesogenesis through direct interaction on host cells [67,68,69]. Despite the lack of studies to correlate certain viral taxonomic groups with obesity in humans and animals, they may have an overlapping spectrum with most persistent viruses associated with obesity (Table 1) [13,14,15,16,17,68,69,70,71].

Evolutionarily, the IFN system represents a signature antiviral mechanism in high vertebrates. The molecular evolution diversifies three types (type I, II, and III) and multiple subtypes/isoforms within type I and type III IFNs. IFN-γ, as the single type II IFN, is mainly produced by NK cells and CTLs and mediates pro-inflammatory response along with the activation of adaptive immune cells. Type I IFNs such as IFN-α/β are expressed by most nucleated cells; in contrast, type III IFNs are produced and active primarily in epithelial cells [11,18,46,51]. Even though type I and III IFNs activate similar downstream signaling cascades in their responsive cells, respectively; type III IFNs (IFNλs), unlike type I IFNs, do not elicit strong inflammatory responses in vivo [72]. The effects in inflammatory induction of type I IFNs are more complicated. It depends on their subtypes, tissue location, and even temporal phases during viral pathogenesis. For example, the universal IFN-α/β subtypes may induce more pro-inflammatory response than IFN-κ or IFN-ε, which have a restrictive expression in keratinocytes or the reproductive tract, respectively [51,71]. In addition, the sustained IFN-α produced during persistent viral infections elicits more inflammatory cytokine responses [11,12]. Nevertheless, most of these antiviral IFN responses were observed during the infection of pathogenic viruses. However, several recent studies discovered that, in addition to the robust IFN reaction to infectious viruses, there is always an intrinsic IFN production signaled by gut microbiota [73,74,75]. In homeostatic situations, gut epithelia and the underlying leukocytes (including NK cells, macrophages, and dendritic cells) sense the natural shedding of microbial molecules from microbiota and sustain an intrinsic expression of innate immune IFNs, particularly IFN-β and probably also the epithelial IFN-λ subtype [72,73,74,75]. Acting through a non-canonical AKT-mTOR pathway (IFN-β) or an IRF1-indolent process (IFN-λ), these intrinsic or local IFNs signal the expression of some anti-inflammatory and cell growth cytokines including IL-10 and IL-13 to maintain an anti-inflammatory microenvironment, which buffers a homeostatic resistance to meta-inflammation in obesity (Figure 1) [76,77,78]. Using hydrodynamic gene transfer to increase peripheral IFN-β1 in mice fed a high-fat diet, a recent animal trial showed that overexpression of *Ifnb1* gene sequentially blocked adipose tissue expansion and body weight gain, independent of food intake [79]. Although more investigations are needed, it is logical to think that the gut virome, especially the “commensal” part should contribute to signaling the intrinsic IFN-β response by microbiota. This thought is largely backed by the role of the gut virome in modulating microbiome composition (such as by phages) as well as more sensitive IFN-inductive effects from viral PAMPs compared to bacterial components [51,77].

## 4. Regulation of Energy and Lipid Metabolism by Interferons 

Instead of directly suppressing or killing viruses, IFNs restrict viruses through induction of hundreds IFN-stimulated effector genes (ISGs) and alter cells to limit virus replication and spreading. These cellular alterations include changes in protein synthesis, energy rebalance, lipid metabolism, membrane composition, and cell proliferation [51,77]. We focused on their role in lipid metabolism to highlight this functional aspect of IFNs because it is plausibly associated with adipogenesis and has been overlooked by most studies of IFN biology in antiviral response [18,80,81,82].

Cell membranes provide a pivotal role and serve as a platform in viral life cycles from entry and replication to assembly and exit. Indeed, the biochemical properties of cell membranes such as fluidity, raft-domain structure, and lipid composition are among the major determinants affecting different stages of viral infections [80,81,82,83,84]. In this context, both cholesterols and sphingolipids (including ceramide) not only affect viral entry, replication, and exit for being structural components that determine the biochemical property of cell membranes, but also actively serve as lipid signaling and direct antiviral molecules to regulate both cellular antiviral responses as well as viral life cycles [83,84]. For example, cholesterol is fundamental for most flavivirus infections in both mammal and insect cell models [85], and 25-hydroxycholesterol (25HC) acts broadly to inhibit viral entry and replication [86]. Similarly, ceramide is an effective sphingosine-derived lipid that regulates diverse cellular immune responses and was recently shown to directly inactivate influenza virus replication [87]. On the other hand, viral infections also extensively modulate (and even actively hijack) cellular lipid metabolism to benefit viral entry and replications. The readers may refer to several recent reviews for details on this topic [83,84,85].

In Figure 2, we illustrate the major effect of type I IFNs in the regulation of cellular energy and lipid metabolism from recent and early studies [88,89,90,91,92,93,94,95,96,97,98,99,100,101,102,103,104,105]. For energy metabolism, type I IFNs enhance glucose uptake in both mouse embryonic fibroblasts and human plasmacytoid dendritic cells (pDCs) [18,88,89,90,91]. Similarly, IFN-induced increases of glycolysis, oxidative phosphorylation, and ATP production derived from the TCA cycle have been observed in multiple human cells including macrophages, DCs, T cells, keratinocytes, and some cancer cell lines [18,80,81,91,92,93,94]. Particularly in macrophages, IFNs also act on the cellular TCA cycle to stimulate the synthesis of reactive oxygen species (ROS) and itaconic acid to augment the cell antimicrobial activity against engulfed pathogens [82,89,95,96,97,98]. The IFN suppression of lipid metabolism is multifaceted, and can occur by directly repressing the synthesis of fatty acid (FA) and cholesterol in HeLa cells, and by regulating the upstream mediators of AMPK and mTOR complex in several human and mouse cells [99,100,101,102,103,104]. Similarly, in livestock cells, we determined that IFNs exert potent antiviral and immunomodulatory activities through interaction with the metabolic system [76]. The expression of IFNs is regulated either by the nutrient sensor mTOR, or by direct reprograming of the lipid metabolic pathways. Thus, a mutual relationship between IFN production and metabolic core processes is manifested in antiviral responses [4,76,99,100,101,102,103]. For example, type I IFNs induce the synthesis of 25-hydroxycholesterol (25HC), a potent metabolite in cholesterol metabolism that exerts a broad antiviral activity and suppressive effect in production of inflammatory cytokines including IL-1 and TNF-α [82,86,96]. In addition, type I IFNs also promote the cross-membrane import of cholesterol and FA into the cells and strengthen membrane rigidity in multiple human cells [80,81]. Potentially, lipogenesis is a sub-effect of the IFN antiviral activity via increase of 25HC synthesis and cholesterol/FA import, particularly under a situation of persistent viral infections. Plasma ceramide and dihydroceramide levels are a biomarker of insulin resistance and obesity, and molecular manipulation of body ceramide to control obesity was illuminated by a recent elegant study [104]. Notably, other reports indicated that both IFN-γ and IFN-β are potent to regulate ceramide metabolism [104,105]. Taken together, the effective IFN response during acute phases of antiviral responses may actually suppress meta-inflammation and reduce energy flow toward lipogenesis in adipocytes. However, the prolonged IFN responses during chronic viral infections might be an adipogenic drive and could lead to immune suppression mostly in obesity patients (see Figure 3 and the next section) [80,81].

## 5. Interferon Responses Underlying the Reciprocal Causality of Obesity and Persistent Viral Infections 

The association of obesity with various viral infections has always faced a “chicken or egg” disputation [13,14,15,16,17]. Do viral infections trigger the infectobesity first, or does obesity lead to higher susceptibility to viral infections that manifest the comorbidity? As described above, we reinforced the reciprocal causality of viral infections and obesity underlying the infectobesity [14]. First, multiple viruses and VLAs have been epidemiologically associated with obesity and some with identified mechanisms in adipogenic or lipogenic induction (Table 1) [13,14,15,16,17]. As more infectious agents beyond viruses are associated with obesity, we propose an immuno-pathological explanation about pathogenic persistence and subverted immune deviation (such as prolonged IFN-α response irritated during chronic viral infections) is a theme underlying viral infectobesity [11,12,16,19]. As depicted in Figure 1, the intrinsic gut IFN-β response and early mucosal IFN-λ response to acute viral infections generally promote less inflammation and cause negligible tissue damage [72,73,74,75]. However, the later and particularly prolonged type I IFN response during the viral persistence entails a systemic inflammatory response in terms of the induction of pro-inflammatory cytokines (including IL-1, IL-6, and IL-18 in myeloid cells and TNF-α from monocytic cells) and activation of inflammatory lineages of macrophages, NK cells, and hepatic T cells [11,12,106,107]. Systemic and adipose tissue-specific meta-inflammation, manifested by these inflammatory cytokines and immune cells, are exaggerated by the immune responses induced by both viral persistence and some obesogens, which sustain obesogenesis and complicate into infectobesity [4,7,64,66].

On the other hand, obesity also provokes an aberration in the immune system including the attenuated and prolonged type I IFN response that shows antiviral inefficacy. Obese individuals have a high level of blood leptin, implying a situation of “leptin resistance”. Leptin potentially upregulates the suppressor of cytokine signaling-3 (SOCS-3). SOCS3 upregulation and altered systemic leptin levels could be responsible for the reduced type I IFN response as well as other immune dysfunction relevant to T cells and B cells in people with obesity [4,14,19,77,106]. IFN signaling occurs via the same JAK–STAT pathway and is negatively regulated by SOCS-3, suggesting a potential mechanism by which IFN responses to viral infections may be suppressed in obese individuals [14,48]. Teran-Cabanillas et al. (2017) have recently evaluated the effect of SOCS expression and type I IFN responses in obese patients, showing that basal SOCS-3 expression was increased in obesity and correlated to the reduction of type I IFN and pro-inflammatory cytokine responses [48]. This obesity-associated immune dysfunction, particularly attenuated IFN responses, resulted in potential increase of host susceptibility to viral infections; therefore leading to increased prognoses of associated viruses from secondary infections, co-infections, and even opportunistic infections in obese patients. A collection of data from clinical cases in humans and studies in mouse models showed that obese individuals exhibited higher susceptibility and mortality during seasonal and epidemic flu infections [5,108]. More severe lung inflammation and lung damage from viral pneumonia as well as prolonged viral shedding were observed in both obese mice and human cases. It was schematically summarized that multiple immune responses including IFN-α/β production and IFN-signaling of ISG production were impaired in the respiratory epithelial cells and macrophages of obese individuals, along with the increase of inflammatory cytokines and M1 polarization of lung macrophages [5,108]. Therefore, the deficiency of IFN production and signaling in obese patients might be among the risk factors for severe outcomes in pandemic influenza infection [5,14,108]. In one report, the incidence of DENV infections seemed higher in obese patients, but needed more epidemiological evidence to justify [20,21]. In the cases of HIV and HCV infections, the persistence of viral infections in obese patients was complicated by accompanied ART or IFN therapies, respectively [22,23,43,44,45,60]; however, evidence from another side suggests that persistent viral infection and perturbation of IFN signaling exacerbate obesity development. Regarding systemic interaction in animal and human bodies, high-fat diet and viral infection (particularly chronic viral infections) may synergistically (or independently sometime) induce dysbiosis of gut microbiota and systemic low-grade inflammation. These systemic changes partially result from attenuated and prolonged IFN production upon persistent viral infections and underlie the systemic inflammation observed in obesity [4,11,12,13,14,15,16,17]. A recent study by Ghazarian et al. (2017) showed that type I IFN–driven activation of CD8+ T cells in the liver correlates with insulin resistance in obesity-linked fat liver (termed nonalcoholic fatty liver disease, NAFLD) in obese patients [107]. This study also critically associated gut microbiome with liver IFN signaling and its role in driving inflammation in the livers of obese mice [107]. It is likely that the dysbiotic microbiome in the obesity and NAFLD patients may harbor persistent infections. In turn, through over-production of long-chain fatty acid (LFA) and leptin by expanding adipocytes, obesity is accompanied with general immune suppression including aberration of antiviral IFN signaling, which leads to high susceptibility to viral infections (Figure 3) [4,11,48,107,109]. Therefore, in most cases of infectobesity, obesity and viral infections may interact dynamically as a reciprocal causality rather than a simple sequential relationship of “chicken or the egg” [14]. Moreover, IFN (particularly type I IFNs) responses have recently been postulated as a central immunological axis that governs adipocyte differentiation and T cell pathogenicity during obesity-associated metabolic disease [107,108,109]. All these observations collectively indicate that IFNs may directly reprogram cellular lipid synthesis and transport, in addition to its indirect effect through interaction with viral infections.

## 6. Prospects of Interferon-Based Anti-Obesity Therapies 

The emerging theme of IFNs as a critical regulator of obesity has been demonstrated by several reports targeting therapeutic development against obesity in both cell and animal models. McGillicuddy et al. (2009) reported that IFN-γ treatment reduced insulin-stimulated glucose uptake in human adipocytes, attenuated insulin sensitivity, and suppressed differentiation of pre-adipocytes to the mature phenotype. The anti-adipogenic effect of IFN-γ is most likely mediated via sustained JAK-STAT1 pathway activation [110]. Using a mouse model, another study detected the stimulation of type I IFN responses by long chain FA in murine hepatocytes and macrophages. Targeting abrogation of IFN-signaling in adipose tissue, but not hepatocyte-specific deletion of Ifnar1 worsened HFD-induced metabolic syndrome. Conversely, improved metabolic control in laparoscopic adjustable gastric banding patients with obesity was associated with stimulation of type I IFN-regulated genes in subcutaneous adipose tissue and liver [111]. This study indicated the differential role of IFN-signaling through the regulation of different sets of ISGs (such as IRFs, IFITs, MX1, and OAS1) in different cells (particularly adipocytes/hepatocytes) that are closely relevant to the development of obesity and metabolic disorders [111]. Again, using hydrodynamic gene transfer to elevate and sustain blood concentration of IFN-β1 in mice suppressed immune cell infiltration and production of pro-inflammatory cytokines, and blocked adipose tissue expansion and body weight gain, independent of food intake [79]. However, such expression may boost a super-physiological and potentially persistent level of IFN-β in the liver. Yet, persistent IFN-β signaling could induce partially immune exhaustion both centrally and locally and would require further study for pharmaceutic safety. Metabolically, IFN-β1, as an intrinsic responder to gut interaction with its microflora (including virome), has been experimentally shown to improve insulin sensitivity and glucose homeostasis in obese mice [46,79]. These findings, even though preliminary, suggest that targeting IFN signaling represents a promising strategy to block obesity development and its related pathologies. So, how about other IFN subtypes besides type II IFN-γ and type I IFN-β? Ying et al. (2014) administered IFN-τ, a ruminant IFN subtype acting as a signaling molecule during pregnancy in trophoblasts, into obese mice induced by HFD [112]. They showed that this livestock IFN subtype not only significantly mitigated obesity-associated systemic insulin resistance and adipose inflammation by controlling macrophage polarization, but also had less cytotoxic effect than typical IFN-β subtype, indicating a therapeutic potential through cross-species screening of IFN molecules [46,79,111,112]. We know that type I IFNs are eminent for their unconventional subtypes, which are functionally prone to anti-inflammatory and reproductive regulation in livestock species such as pigs and cattle as well as type III IFNs that are restricted to epithelial expression [46]. Information from the study of tissue-specific IFN-signaling effect by Wiser et al. (2018) implies that epithelial IFN-signaling has limited regulation on metabolic and hepatic diseases [111]. Based on these insightful studies for IFN-based anti-obesity therapies, a cross-species and family-wide screening of the most effective IFN ligands is needed, and may provide more IFN-based options for therapeutic optimization against current obesity prevalence and reveal IFN subtypes that bear superior activity in anti-obesity or metabolic regulation to IFN-β or IFN-γ [46,79,110,111,112].

## 7. Conclusive Remarks 

Interferons, as a family of key cytokines, exert broad antiviral, anti-tumor, and anti-proliferative activities in vitro and in vivo [11,18,46,51]. The broad activity of IFNs depends on inducing hundreds of ISGs, which not only directly restrict virus infections at different stages of viral life cycle, but also orchestrate cell metabolism to enter into an antiviral state [11,51]. In this manner, IFNs have been found to play a role in reprogramming cell lipid metabolism in addition to their activity in coordinating RNA and protein metabolism [80,81,88]. This underpins a metabolic basis of IFNs in the development of obesity, and particularly, the infectobesity involving viral infections [13,14,15,16,17]. Effective IFN responses, especially those mediated by type I and type III IFNs, are postulated to be an immunological axis that determines antiviral defense. However, the futile and prolonged IFN response accompanying persistent viral infections actually compromise body function in both antiviral immunity and metabolic normality [11,12]. On the other hand, the cross-talking of leptin and LFA with IFN signaling provides another rationale to associate IFNs with obesity [29,48]. However, more direct evidence such as using IFN-signaling deficient animal models are needed to conclusively determine if and how fundamental IFN signaling plays a role in obesity development. Remarkably, all immune responses are a double-edged sword, as it is for IFN-signaling [11,12,19,50,51]. Although the intrinsic and properly stimulated IFN responses are favorable for antiviral and anti-obesity effect; on the other hand, there are multiple literatures also suggesting that improper IFN responses may actually impair viral clearance, and promote hepatic and metabolic diseases [19,50,51,107]. Thus, for the development of IFN-based therapeutics for anti-obesity purposes, effective IFN regimens need to be selected and tested dynamically to overcome potential side-effects such as those manifested from antiviral IFN therapeutics [113].

## Figures and Tables

**Figure 1 biomolecules-09-00726-f001:**
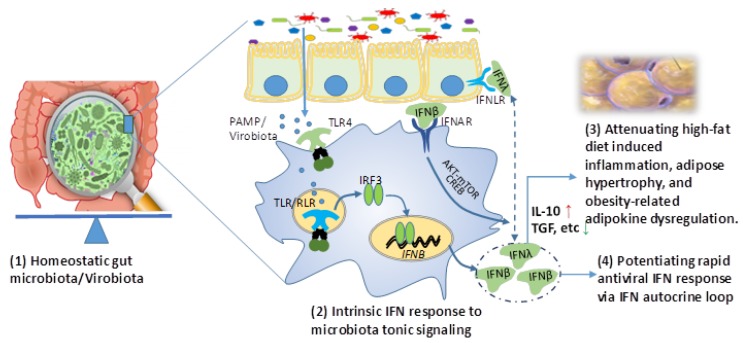
Intrinsic interferon (IFN) response to microbiota tonic induction may suppress obesity in addition to its role in potentiating rapid antiviral response. In homeostatic situation, gut epithelia and the underlying leukocytes (mainly macrophages and dendritic cells, for example) sense natural shedding of microbial molecules from microbiota (particularly virobiota) and sustain an intrinsic expression of innate immune IFNs, particularly including IFN-β and probably IFN-λ. Acting through a non-canonical AKT-mTOR pathway, intrinsic IFNs signal IL-10/TGF production and contribute to maintain an anti-inflammatory microenvironment, which attenuates meta-inflammation and adipogenesis related to visceral obesity. The anti-obesity effect of transfecting IFN-β was recently demonstrated in addition to its role in potentiating rapid antiviral response via IFN autocrine loop of regulation. Abbreviations: AKT-mTOR, protein kinase B and mammalian target of rapamycin pathway; CREB, cAMP response element binding protein; IFN, interferon; IFNAR, type I IFN receptor; IFNLR, type III IFN receptor; IRF, IFN regulatory factor; IL-10, interleukin 10; PAMP, pathogenic associated molecular pattern; TGF, transforming growth factor; and TLR, Toll-like receptor.

**Figure 2 biomolecules-09-00726-f002:**
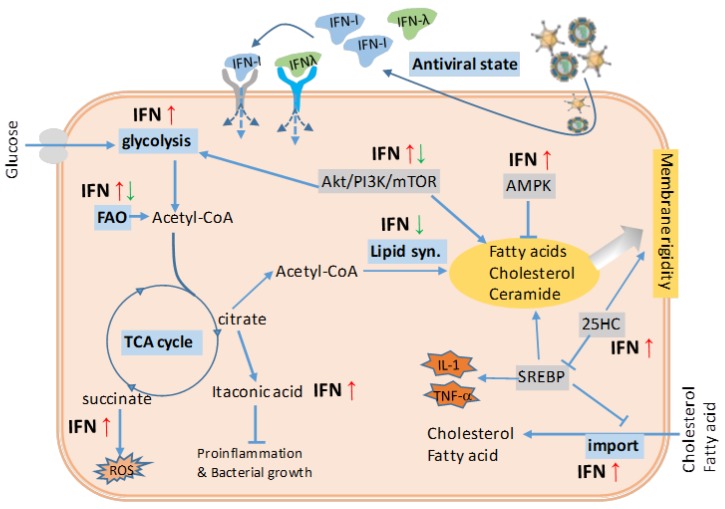
Antiviral IFN stimulation and the effect on energy and lipid metabolism. Counteracting viral infections, which seize the cell energy and lipid metabolism for viral production but enhance fatty acid (FA) and cholesterol synthesis, effective IFN response leads to an antiviral state through an IFN autocrine or paracrine regulatory loop. The induction of antiviral state is accompanied by a general arrest of protein and lipid metabolism. The IFN suppression of lipid metabolism is multifaceted such as directly repressing the synthesis of FA and cholesterol, and regulating via the upstream regulators of AMPK and mTOR complex. In addition, IFNs have been also implicated in the regulation of glycolysis, TCA cycle, and cholesterol transport to facilitate cell antiviral response. Taking together, the effective IFN response, especially that during acute phases of antiviral responses, is unlikely to induce obesity and meta-inflammation underlying obesity; however, the prolonged IFN responses during chronic viral infections might be an adipogenic drive in an opposite way (Figure 3). Abbreviations: acetyl-CoA, acetyl coenzyme A; Akt/PI3K/mTOR, phosphatidylinositol-3-kinase (PI3K)/Akt and the mammalian target of rapamycin (mTOR) signaling pathway; AMPK, 5-adenosine monophosphate-activated protein kinase; FA, fatty acid; FAO, fatty acid oxidation; IFN-I, type I interferon; 25HC, 25-hydroxycholesterol; ROS, reactive oxygen species; SREBP, sterol regulatory binding protein; TCA cycle, tricarboxylic acid cycle. Adapted from [80,81] with a permission.

**Figure 3 biomolecules-09-00726-f003:**
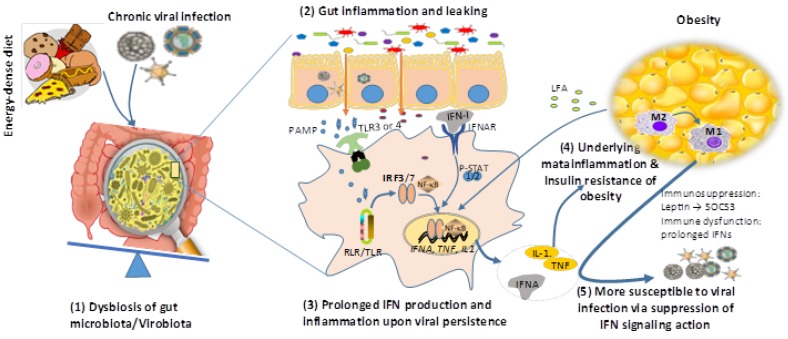
A reciprocal causality of obesity and chronic viral infections. High-fat diet and viral infection (particularly chronic viral infections) may independently (or synergistically in most time) induce dysbiosis of gut microbiota leading to gut inflammation and leaking, which at least partly results from prolonged IFN production upon chronic viral infection and underlies systemic inflammation of obesity. In turn, through the production of long-chain fatty acid (LFA) and leptin by expanding adipocytes, obesity is accompanied with meta-inflammation (enhanced by LFA) and susceptibility to viral infections (through suppression of IFN antiviral signaling by leptin induction of SOCS3 signaling). Abbreviations: NF-κB, nuclear factor-kappa B; P-STAT, phosphorylated signal transducer and activator of transcription; RIG-I, retinoic acid-inducible gene I; SOCS3, suppressor of cytokine signaling 3; TNF, tumor necrosis factor; and that in the legend of Figure 1.

**Table 1 biomolecules-09-00726-t001:** Adipogenic viruses or virus-like agents and virus-interferon (IFN) interaction *.

Adipogenic Viruses *	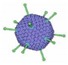 Adenoviridae [15,16,17,18,19,20,21,22,23,24,25,26,27,28,29,30]	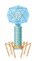 Gut Phages [31,32,33,34]	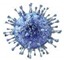 Herpesviridae [35,36]	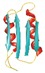 Slow Virus (Prion) [37,38,39,40,41,42]	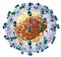 Other Viruses [21,22,23,43,44,45]
Natural hosts orLivestock reservoirs	Avian: SMAM1 Human:Ad-5/9/31/36/37	Gut phages likely in all animals	Human:HSV-1, CMV, HHV8	Sheep: ScrapieCattle: BSE/CJDHuman: Kuru	Avian: RAV7Sheep/Horse: BDVHuman: DENV, HIV, HCV
Lipogenic in vitro	Ad-5/9/31/36/37Cause adipocyte differen-tiation and lipogenesis	Unknown	Increase lipogenesis in cells	Unknown	HCV enhances lipidsynthesis, and CDVenlarges adipocytes
Adipogenic in animals	Ad-36, Ad-37: chickensAd-36, Ad-5: miceAd-36: rats, marmosets	Gut phages followingrisperidone treatment: mice	Unknown	BSE: primatesScrapie, CJDvariants: mice	BDV, CDV: miceRAV7: chickens
Obesity-association in humans	Ad-5: childhood obesity, Ad-36: childhood, adult obesity and BMISMAM-1: BMI	Adipogenic gutmicrobe transferassociated with heavierhuman donor	CMV: metabolicSyndromeHSV-1: adult obesity	Kuru obesity and/orbulimia duringearly disease inhumans	HCV genotype 3:insulin resistance in humansDENV: children obesityHIV: obesity in patients on ART
Obesity during persistent viral infections? [13,14,15]	YES	Unknown, but the effect of gut phages on microbiota is persistent	YES	YES	YES, especially in human cases
Suppression of acute IFN antiviral response, and [43,44,45,46,47,48,49,50,51,52,53,54,55,56,57,58,59,60,61,62,63]	Yes, through E1A gene	Unknown	Yes, HSV-1 through miRNA; and CMV has multiple IFN-antagonistic mechanisms	Infecting prions suppress interferon expression	Multiple IFN-antagonistic mechanism in DENV, HIV, and HCV
Cause prolonged IFN production upon chronic infection [43,44,45,46,47,48,49,50,51,52,53,54,55,56,57,58,59,60,61,62,63]	Likely yes in adipogenic Ad-36 infection, and persistent enteric infection in children	Expansion of gut phages induces intestinal inflammation via TLR9-mediated IFN reaction	CMV exploits prolonged IFN production to induce IL-27 production and chronic infection	Typically chronic, as reflected by the term of “slow virus”	Prolonged IFNs promote HCV and retroviral chronicity.

*****Abbreviations: Ad, adenovirus; ART, HIV Antiretroviral therapy; BDV, borna disease virus; BMI, body mass index; CMV, cytomegalovirus; DENV, dengue fever virus; HHV, human herpesvirus; HCV, hepatitis C virus; HSV-1, herpes simplex virus 1; IL, interleukin; RAV, Rous-associated virus; SMAM, an avian adenovirus from India; TLR9, Toll-like receptor 9. Contents in Row 1–4 adapted from Voss and Dhurandhar (2017) [15].

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
