# Peer review of "Viral Infections and Interferons in the Development of Obesity"

_biomolecules, 2019, doi:10.3390/biom9110726_

Round 1

Reviewer 1 Report

The authors have addressed the majority of my concerns. The only remaining point I would make (that stems from responses to my previous review) is to change the title to more accurately reflect that the paper is more about the role of infections in the development of obesity versus host-pathogen interactions in the context of pathogenesis of viral infection in the obese host.

Author Response

(2nd revision) Point-to-point response to the reviewers’ comments:

Reviewer 1:

The authors have addressed the majority of my concerns. The only remaining point I would make (that stems from responses to my previous review) is to change the title to more accurately reflect that the paper is more about the role of infections in the development of obesity versus host-pathogen interactions in the context of pathogenesis of viral infection in the obese host.

We agree and the title was changed to: “Viral infections and Interferons in the Development of Obesity”

Reviewer 2:

The review article titled “Viral Infections and Obesity: Interferons Come into Play” by Tian et al., attempts to summarize the role of interferons in the context of obesity. This review article is interesting and important considering the relevance of IFNs in the etiology of infectobesity. Despite the fact that many groups have now shown that there is a link between viral infections and obesity, a clear cause-effect relationship has not been reached in this field and this article provides a good review of the existing literature on this topic. The authors provide a comprehensive review on existing literature while maintaining a balanced standpoint in interpreting results/findings from other groups.

We appreciate for this positive comment.

Following are some minor issues that needs to be fixed to improve the overall quality of the review:

Table 1: the content is cut. Unable to view the last column that talks about Prions.

Table 1 has been re-incorporated and confirmed for complete demonstration in the PDF file

Throughout the article, Abbreviations are inconsistent: Some of them are misspelled: TGF beta-Transforming growth factor and not “tissue” growth factor. In some places, the abbreviations are not given: LAGB (line # 386) In Figures 1,2 and 3: IFNAR and IFNLR are two different receptors that act as cognate receptors for type1 and typeIII IFNs respectively. However, the diagram seems to depict both IFNAR and IFNLR as the same. This is scientifically misleading and inaccurate particularly for readers who do not work in this field. Hence this needs to be corrected.

We had an extensive proofreading to make sure the consistence about Abbreviations, and corrected the typos.

Reviewer 2 Report

The review article titled “Viral Infections and Obesity: Interferons Come into Play” by Tian et al., attempts to summarize the role of interferons in the context of obesity. This review article is interesting and important considering the relevance of IFNs in the etiology of infectobesity. Despite the fact that many groups have now shown that there is a link between viral infections and obesity, a clear cause-effect relationship has not been reached in this field and this article provides a good review of the existing literature on this topic. The authors provide a comprehensive review on existing literature while maintaining a balanced standpoint in interpreting results/findings from other groups.

Following are some minor issues that needs to be fixed to improve the overall quality of the review:

Table 1: the content is cut. Unable to view the last column that talks about Prions. Throughout the article, Abbreviations are inconsistent: Some of them are misspelled: TGF beta-Transforming growth factor and not “tissue” growth factor. In some places, the abbreviations are not given: LAGB (line # 386) In Figures 1,2 and 3: IFNAR and IFNLR are two different receptors that act as cognate receptors for type1 and typeIII IFNs respectively. However, the diagram seems to depict both IFNAR and IFNLR as the same. This is scientifically misleading and inaccurate particularly for readers who do not work in this field. Hence this needs to be corrected.

Author Response

(The authors gave the same response as above.)
